# Hyperglycaemia-Induced Downregulation in Expression of nNOS Intramural Neurons of the Small Intestine in the Pig

**DOI:** 10.3390/ijms20071681

**Published:** 2019-04-04

**Authors:** Michał Bulc, Katarzyna Palus, Michał Dąbrowski, Jarosław Całka

**Affiliations:** 1Department of Clinical Physiology Faculty of Veterinary Medicine, University of Warmia and Mazury, Oczapowskiego Str. 13, 10-719 Olsztyn, Poland; katarzyna.palus@uwm.edu.pl (K.P.); jaroslaw.calka@uwm.edu.pl (J.C.); 2Department of Veterinary Prevention and Feed Hygiene, Faculty of Veterinary Medicine, University of Warmia and Mazury in Olsztyn, Oczapowskiego Str. 13, 10-719 Olsztyn, Poland; michal.dabrowski@uwm.edu.pl

**Keywords:** hyperglycaemia, nitric oxide synthase, enteric nervous system, pig

## Abstract

Diabetic autonomic peripheral neuropathy (PN) involves a broad spectrum of organs. One of them is the gastrointestinal (GI) tract. The molecular mechanisms underlying the pathogenesis of digestive complications are not yet fully understood. Digestion is controlled by the central nervous system (CNS) and the enteric nervous system (ENS) within the wall of the GI tract. Enteric neurons exert regulatory effects due to the many biologically active substances secreted and released by enteric nervous system (ENS) structures. These include nitric oxide (NO), produced by the neural nitric oxide synthase enzyme (nNOS). It is a very important inhibitory factor, necessary for smooth muscle relaxation. Moreover, it was noted that nitrergic innervation can undergo adaptive changes during pathological processes. Additionally, nitrergic neurons function may be regulated through the synthesis of other active neuropeptides. Therefore, in the present study, using the immunofluorescence technique, we first examined the influence of hyperglycemia on the NOS- containing neurons in the porcine small intestine and secondly the co-localization of nNOS with vasoactive intestinal polypeptide (VIP), galanin (GAL) and substance P (SP) in all plexuses studied. Following chronic hyperglycaemia, we observed a reduction in the number of the NOS-positive neurons in all intestinal segments studied, as well as an increased in investigated substances in nNOS positive neurons. This observation confirmed that diabetic hyperglycaemia can cause changes in the neurochemical characteristics of enteric neurons, which can lead to numerous disturbances in gastrointestinal tract functions. Moreover, can be the basis of an elaboration of these peptides analogues utilized as therapeutic agents in the treatment of GI complications.

## 1. Introduction

Diabetes mellitus is a complex disease resulting in altered glucose metabolism. In both type 1 and type 2 diabetes, pancreatic β cells cannot secrete enough insulin to maintain an appropriate blood glucose level. Thisleads to long-term episodes of hyperglycaemia. The central and peripheral neurons are also highly sensitive to increased glucose levels. It is known that hyperglycaemia permanently disturbs GI tract function. Indeed, up to 75% of patients with diabetes complain of post-prandial fullness with nausea, bloating, abdominal pain, diarrhoea and/or constipation [1,2,3]. On the other hand, due to the large number of neurons in the alimentary tract and its fundamental role in theregulation of digestive processes, it is worth understanding the response of the ENS neurons to hyperglycaemia. It needs to be highlighted that gastrointestinal autonomic neuropathy may apply toany organ in the digestive tract, such as the oesophagus, stomach, pancreas and the small and large intestines.

It is well known that the enteric nervous system is responsible for the control of the gastrointestinal (GI) tract functions. The ENS neurons are present in all segments of the GI tract from the oesophagus to the anus [4]. They are highly organized and grouped in the intramural ganglionated plexuses. The types and numbers of these plexuses strongly depend on the animal species as well as the part of gastrointestinal tract studied [5,6,7]. In large animals, such as ruminants and pigs, as well as humans, the ENS is organized into two plexuses inside the stomach: (1) the myenteric plexus (MP), situated between the longitudinal and circular muscle layers, and (2) the submucous plexus (SP) located close to the lamina propria of the mucosa. However, in the small and large bowel, the SP is divided into outer submucous plexus (OSP) (located near to the external circular muscle layer) and inner submucous plexus (ISP) (situated between the muscularis mucosa and lamina propria) [1,4,6]. The ENS neurons participate in the control and regulation of most of the GI functions [8]. Among the most important are the motor activity of each segment of GI, blood flow, secretion of gastric fluids, absorption of food content, as well as regulation of immune response (together with an immune cellular competent) [9,10]. In addition, the morphological and functional features of enteric neurons are also diverse in their neurochemistry. To date, many biologically active substances utilized by enteric neurons have been described [11]. It is well-known that each neuron can synthesize and secrete more than one active substance. The most important and best-known representative of this class of substances is nitric oxide [12,13].

Nitric oxide is an unstable gas that has the features of a free radical. It is synthesized in the body from L- arginine under the control of a specific enzyme nitric oxide synthase (NOS) [14]. Inside the enteric neurons, nitric oxide (NO) is considered as a specific marker of nitrergic neurons [15]. To date, nitric oxide synthase inside enteric neurons has been described in pigs, guinea pigs, rodents and humans [6,16,17,18]. In the gastrointestinal tract, nitric oxide is an inhibitory neurotransmitter. The inhibitory function of NO is particularly visible in reference to the smooth muscle of the digestive tract, where it causes relaxation of smooth muscle and decreases the motor activity of the GI tract. Moreover, the secretion of enteric hormones and fluids is also reduced under the influence of nitrergic signalling. Another function of NO is the regulation of mesenteric and intestinal blood flow. In this case, nitric oxide seems to play a vasodilatory role [14]. The adaptation of neurons to the pathological factors is expressed by changes in expression (decreased or/and increased) of neuroactive substances. Some of these substances are assigned a neuroprotective role and thus protect neurons from apoptosis. Previous results have indicated the neuroprotective properties of nitric oxide in enteric neurons [19,20]. 

In recent years, considerable effort has been devoted to the elucidation of the molecular mechanisms of gastric complications during diabetes. Therefore, the goal of this study was to provide novel data on the influence of streptozotocin (STZ)-induced hyperglycaemia on the nitrergic neurons in the ENS of the porcine small intestine. Moreover, among the numerous diabetes complications especially inside nervous structures the pathomechanism of ENS function impairment is relatively poorly understood. Only full understanding of this mechanism and understanding the role of various biological factors including neurotransmitters will allow to conduct research on substances with a potential therapeutic effect. Therefore, the choice of the pig as an animal model was not incidental. The pig has obtained notable importance, as its metabolic and pathophysiological responses to the pathological factors partly mimic those observed in humans [21,22]. It has been shown that the blood supply of the pig pancreas is similar to that of humans and the number of β-cells is also within a similar range, making it a valuable model for the study of diabetes [23]. The most accepted porcine model of diabetes is STZ-induced type 1 diabetes, where glucose and insulin secretion remain at a very similar level to human diabetic patients [2,24]. Moreover, throughout the experiment, a high glucose level was noted. To date, the type 1 STZ-induced hyperglycaemic/diabetic pig model has been previously used in studies of the cardiovascular complications of the disease, while neuronal complications, especially in the GI tract in this model, are still unclear [14,25].

## 2. Results

All animals (in both the control and experimental groups) survived the experiment in good general condition. Pigs thatreceived streptozotocin had similar appetite, body weight and general health to animals in the control group. Moreover, none of the animals required exogenous insulin supplementation.

### 2.1. Glucose Serum Level

The mean glycaemia before diabetes induction in all animals was within standard reference values for the pig (5.01 mmol/L ± 0.10 mmol/L) (Table 1). After streptozotocin injection, systematic growth of the mean glucose level was observed. Starting from the first week of experiment, the baseline glucose level in experimental animals remained at a level exceeding 20 mmol/L. All results of the glucose level are presented in Table 1. 

### 2.2. Changes in Chemical Coding of the ENS Neurons in the Small Intestine

Experimentally-induced hyperglycaemia resulted in alternations in the percentage of enteric neurons immunoreactive to the investigated substances. The character and intensity of these changes clearly depended on the type of enteric plexus and active substance studied, as well as on the parts of the intestine under investigation. 

#### 2.2.1. Myenteric Plexus (MP)

##### Neuronal Nitric Oxide Synthase (nNOS) Activity

In the control group, nNOS-immunoreactive (nNOS-IR) neurons were detected in all segments of the small bowel (Figure 1). The highest number of nNOS-positive cell bodies in comparison with all Hu C/D—positive cells was present in the jejunum (24.43% ± 2.09) (Figure 2B). A slightly smaller population nNOS-containing neurons (22.62% ± 0.56) was detected in the duodenum (Figure 2A), although in the ileum nNOS-IR perikarya constituted 18.44% ± 0.22 of all MP neurons (Figure 2C). Following diabetes induction, a statistically significant decrease in nNOS-positive neurons was noted (Figure 1). The decrease was reported in all segments of the small intestine. In the MP of duodenum, we noted that 14.44% ± 0.53 of neurons showed nNOS expression (Figure 2D), a similar number was visible in the ileum 13.21% ± 0.59 (Figure 2F), whereas the largest drop was reported in the jejunum (10.57% ± 0.47) (Figure 2E). 

##### Co-Localization of nNOS with Other Biologically Active Substances in the MP

In the present study, the co-localization of nNOS with vasoactive intestinal polypeptide (VIP), galanin (GAL) and substance P (SP) were observed both in the control animals and following the administration of the streptozotocin (Figure 3, Figure 4, Figure 5, Figure 6, Figure 7 and Figure 8). The degree of co-localization of nNOS with the above-mentioned substances depended on the part of the small bowel. The degree of co-localization with VIP (Figure 3) in the control group was similar in all parts of small intestine and amounted to 20.95% ± 0.43 in the duodenum (Figure 4A), 21.62% ± 0.65 in the jejunum (Figure 4B) and 20.64% ± 0.22 in the ileum (Figure 4C), respectively. In the experimental group, a statistically significant increase in VIP expression in nNOS-containing neurons in the MP of the duodenum (31.59% ± 0.54) (Figure 4D) and jejunum (31.62% ± 0.41) (Figure 4B) was noted, while in the ileum the changes were not statistically significant. Another substance that was observed in nNOS-IR neurons was GAL (Figure 5). The degree of co-localization of these two neuronal factors under physiological conditions was estimated in the duodenum, on a similar level as VIP (22.89% ± 0.44) (Figure 6A). However, in the jejunum and ileum, the percentage of co-localization with GAL was higher than in the VIP and was 28.20% ± 0.70 in the jejunum (Figure 6B) and 31.29% ± 0.37 in the ileum (Figure 6C), respectively. Under hyperglycaemia conditions, we observed an increase in GAL expression within nNOS-containing cell bodies. The degree of augmentation depended on the parts of the intestine. A higher increase was visible in the ileum (49.66% ± 0.51) (Figure 6F) and jejunum 42.48% ± 0.95 (Figure 6E). In turn, in the duodenum the co-localization was estimated at 37.78% ± 0.37 (Figure 6A). The last substance investigated in the present study was SP (Figure 7). In the control pigs, the smallest degree of co-localization with nNOS was observed in the MP of duodenum (19.21% ± 0.32) (Figure 8A), while in the jejunum and ileum, the degree of co-localization with nNOS was similar in the jejunum (30.96% ± 0.12) (Figure 8B) and 29.38% ± 0.61 in the ileum (Figure 8C). In the experimental animals, we noted an increase in SP expression. In the duodenum, the number of nNOS-IR neurons that simultaneously expressed SP amounted to 42.18% ± 0.69 (Figure 8D). A similar number were visible in the jejunum (43.25% ± 0.79) (Figure 8E) and the ileum (43.01% ± 0.68) (Figure 8F). 

#### 2.2.2. Outer Submucosal Plexus (OSP) 

##### Neuronal Nitric Oxide Synthase (nNOS) Activity

Under physiological conditions in the duodenum, nNOS positive neurons were estimated at 20.59% ± 0.62 (Figure 2G), in the jejunum, the number of nNOS neurons amounted to 20.25% ± 0.60 (Figure 2H), while in the ileum 15.95% ± 0.11 (Figure 2I) of all Hu C/D neurons contained nNOS. In the experimental group, as in other plexuses, a decrease in nNOS-IR neurons was noted. In the duodenum, the number of nNOS neurons was estimated at 16.56% ± 0.53 (Figure 2J), while in the jejunum it was 11.00% ± 0.59 (Figure 2K) and in the ileum it was 12.07% ± 0.67 (Figure 2L), respectively. 

##### Co-Localization of nNOS with Other Biologically Active Substances in the OSP

In the control group, the percentage of VIP-IR cell bodies within the nNOS-IR neurons was estimated at 16.29% ± 0.76 in the duodenum (Figure 4G), 20.38% ± 0.82 in the jejunum (Figure 4H) and 11.74% ± 0.51 in the ileum (Figure 4I). However, under experimental conditions the number of VIP-containing neurons increased in all segments of the small intestine, reaching the following values: 20.55% ± 0.27 in the duodenum (Figure 4J), 30.07% ± 0.25 in the jejunum (Figure 4K) and 16.25% ± 0.63 in the ileum (Figure 4L). In turn, GAL-expressing neurons in the control group in relation to nNOS-IR were estimated at 15.32% ± 0.29 in the duodenum (Figure 6G), 11.46% ± 0.26 in the jejunum (Figure 6H) and 21.93% ± 0.38 in the ileum (Figure 6I). Following streptozotocin injection, the increase in GAL expression was noted as 22.41% ± 0.75 in the duodenum (Figure 6J), 21.20% ± 0.46 in the jejunum (Figure 6K) and 31.53% ± 0.53 in the ileum (Figure 6L). The neurons simultaneously expressing SP and nNOS in the control group constituted 41.21% ± 0.35 in the duodenum (Figure 8G), 10.41% ± 0.30 in the jejunum (Figure 8H) and 26.49% ± 0.27 in the ileum (Figure 8I). Under experimental conditions, the number of SP-IR neurons increased. In the duodenum, SP-IR/nNOS-IR neurons were estimated at 17.65% ± 0.42 (Figure 8J), while in the jejunum SP-IR/nNOS-IR cells reached 19.20% ± 0.69 (Figure 8K), and 31.46% ± 0.63 in the ileum (Figure 8L).

#### 2.2.3. Inner Submucous Plexus (ISP)

##### Neuronal Nitric Oxide Synthase (nNOS) Activity

In the control group, nNOS neurons accounted for 16.02% ± 0.52 of all ISP neuronal cells in the duodenum (Figure 2M), 25.35% ± 0.89 in the jejunum (Figure 2N) and 12.23% ± 0.33 in the ileum (Figure 2O). Under diabetic conditions, only nNOS-positive neurons of ISP in the duodenum did not show statistically significant changes, while in the jejunum and ileum we noted a decreased number of nNOS- IR 18.09% ± 0.47 in the jejunum (Figure 2R) and 9.50% ± 0.43 in the ileum (Figure 2S) were noted. 

##### Co-Localization of nNOS with Other Biologically Active Substances in the ISP

The percentage of nNOS-IR and VIP-IR perikarya in the control pigs was estimated at 17.24% ±0.41 in the duodenum (Figure 4M), 20.78% ± 0.82 in the jejunum (Figure 4N) and 10.42% ± 0.40 in the ileum (Figure 4O). During hyperglycaemia conditions, a clearly visible increase in VIP expression was observed in all investigated areas of the small intestine (Figure 3). The exact values were as follows: 20.27% ± 0.35 in the duodenum (Figure 2P), 25.89% ± 0.85 in the jejunum (Figure 4R) and 15.05% ± 0.23 in the ileum (Figure 4S). The percentage of GAL in relation to nNOS in the ISP under physiological conditions was relatively small in the jejunum 8.95% ± 0.53 (Figure 6N), while a slightly higher number was present in the duodenum (13.88% ± 0.46) (Figure 6M) and ileum (19.76% ± 0.24) (Figure 6O). In experimental animals, we observed an increase in GAL expression within all segments of the small bowel. In the duodenum, the percentage of GAL-IR cells in the population of nNOS-positive neurons was 20.00 % ± 0.47 (Figure 6P), in the jejunum it was 15.03 % ± 0.49 (Figure 6R) and in the ileum it was 34.05% ± 0.90 (Figure 6S). In addition, SP was present in nNOS-positive neurons. In the control group in the duodenum we observed that 20.24 % ± 0.46 nNOS-IR cell bodies contained SP (Figure 8M), while in the jejunum their number was smaller (8.60% ± 0.71) (Figure 8N), whereas in the ileum 26.26% ± 0.61 of nNOS-IR neurons also contained SP (Figure 8O). In the experimental group in the duodenum we did not observe statistically significant changes, while in the jejunum (16.92 % ± 0.50) (Figure 8R) and ileum (39.81 % ± 0.36) (Figure 8S) an increase in SP expression was present, respectively. 

## 3. Discussion

The present study demonstrated that porcine enteric neurons located inside the small intestine after six weeks of permanent hyperglycemia induced by streptozotocin injection exhibit variability in the number of nNOS positive neurons. Moreover, for the first-time co-localization of nNOS with other biologically-active substances in animals with high glucose levels was described. As shown in previous studies, nitric oxide is a commonly occurring neurotransmitter within different parts of gastrointestinal tract [5,26,27,28]. Interspecies differences in the localization of nNOS-positive neurons inside the gastrointestinal tract were observed, both in each plexus as well as particular fragments of digestive tract. The presence of NOS and NADPH diaphorase- positive neurons was previously described in pigs [6,17], rats [29], mice [30], cats [31] and humans [32]. Moreover, this study focused on exact characterization of all enteric plexuses the gut and we found nNOS-positive neurons were found in both the submucosal plexuses (outer and inner) and in myenteric neurons. In the current study, the largest number of NOS-positive neurons was observed in the myenteric plexus, the outer submucosal plexus contained slightly fewer NOS-immunoreactive neurons, while the inner submucosal plexus exhibited the smallest quantity of NOS-containing neurons. In the available literature, there are few results concerning the distribution of nitrergic nervous structures in the porcine ENS, especially in the small intestine [5,33]. The current results are congruent with a previous study conducted in pigs [6] in which, as in the present study, more numerous NOS-positive neurons were found in the myenteric plexus within the small intestine [5]. However, a similar relationship does not occur in other segments of gastrointestinal tract. For example, in the porcine descending colon, a large population of nNOS-LI neurons were present in the inner submucosal plexus [6]. In addition, in the guinea pig, particularly numerous neurons exhibiting the presence of nNOS were noted in the descending colon [31]. This discrepancy with respect to species and segments of the GI tract showing specific differences of nNOS distribution, highlight the diversity of the gaseous neurotransmitter in the digestive tract. One of the most important functions of nitric oxide in the GI tract is a reduction of the smooth muscle layer contractile activity within its various parts. This leads to a consequent decrease in motor activity. Moreover, nitric oxide, by regulation of blood flow, affects mucosal and mesenteric perfusion [25]. In GI smooth muscles, the NO-induced relaxation is mediated by the guanylyl cyclase-cyclic GMP pathway and by cell hyperpolarization [34]. Other important functions of NO that are also associated with the regulation of blood flow include the regulation of salt and water secretion and resorption [35]. 

During the present study, we found a relationship between nitrergic innervation of the gut and hyperglycaemia, as well as the influence of diabetes on the enteric neurons with particular emphasis on different parts of small intestine. Insufficient supply of insulin occurring in the course of diabetes leads to hyperglycaemia and the development of diabetes’ complications. It is well-known that augmentation of glucose level disrupts normal functions of neurons [36]. The consequences of chronic hyperglycaemia is, among others, the development of gastrointestinal autonomic neuropathy. This pathology may affect each organ in the digestive tract, such as the oesophagus, stomach or small and large intestine [37]. In recent years, considerable effort has been devoted to the elucidation of the molecular mechanisms underlying pathogenic diabetic gastrointestinal neuropathy. The current results shows that, experimentally-induced hyperglycaemia led to a decrease in the population of nNOS-positive neurons, particularly in the myenteric plexus. In turn, the number of nNOS-expressing neurons inside the inner and outer submucosal plexus was also decreased, but to a minor extent than in the myenteric plexus. The finding of decreased numbers of nitrergic neurons confirms previous observations in spontaneous and induced animal models of type 1 diabetes [38,39]. Rodent models of diabetes indicate that in the initial stage of diabetes, there is a loss of nNOS content and function, however, at later stages there is nitrergic degeneration with a complete loss of nitrergic function [39]. Altogether, hyperglycemia seems to be associated with a reduction in nNOS- positive neurons within myenteric neurons. On the other hand, we observed a slight increase in NOS-positive neurons within submucosal plexuses. This may indicate a different function of the submucosal plexuses in intestinal pathophysiology compared to myenteric plexuses. 

The obtained results raise the question of how chronic hyperglycaemia affects the obtained changes in small intestine enteric neurons. The long-term persistence of elevated levels of glucose in the blood also results in an increase in the intracellular concentrations of glucose that can lead to a number of metabolic changes [40]. One of them is the initiation of non-enzymatic glyco-oxidation of numerous proteins. This multi-stage process also leads to the formation of advanced glycation end-products (AGEs). Moreover, enhanced protein glycation leads to activation of the signal transduction receptor, RAGE (Receptor for Advanced End-Glycation products) [41]. Additionally, glucose itself can undergo autoxidation, forming reactive oxygen species [42]. In turn, activation of this receptor leads to significant biochemical changes in the cells. First of all, there is a shift in the balance between the production of reactive oxygen species (ROS) and the antioxidant defense systems, which results in the development of oxidative stress. Secondly, via the up-regulation of the NF-κB pathway, pro-inflammatory protein synthesis is increased [43] resulting in reduced nNOS dimerization [18]. In turn, dimerization is required for the enzymatic activity of NOS. This ultimately leads to a decline in the population of NOS positive neurons. However, it should be emphasized that the decrease observed in the present study in the number of NOS-positive neurons is not persistent. For this reason, it has been suggested that dysfunction of nitrergic transmission is reversible [39]. There are studies with animal streptozotocin-induced diabetes where treatment with insulin restores nNOS synthesis [44]. Overall, diabetes with long term episodes of hyperglycaemia seems to be associated with a reduction in the number of nNOS-positive neurons in intramural ganglia with a possible regenerative response at later stages when there is compensation of the glycaemia level. Evidently, these changes may contribute to the gastric motor activity disturbances in diabetes. Previous experimental studies, as well as clinical observation of people with diabetes, indicated a harmful effect of high glucose level on the gastrointestinal function [43]. One of the more serious disorders is delayed gastric emptying, constipation and postprandial fullness [37]. Loss of function of NOS- containing neurons has a significant influence on these disorders. Pharmacological inhibition of NOS by a three-day oral administration of L-NG-nitroarginine methyl ester results in delayed gastric emptying [44]. In the small bowel, a decrease in nitric-expressing neurons during diabetes leads to a decrease in phase 3, which results in the prolongation of migrating myoelectric complex cycle [45]. Our results also confirm that in the swine diabetic model, hyperglycaemia significantly reduced the number of NOS-containing neurons in the small intestine. 

Although nitrergic innervation represents non-cholinergic and non-adrenergic components, it is well-known that NOS- positive neurons may express more than one active substance [6,7]. The use of fluorescence microscopy as well as double and triple staining techniques has proved that each neuron synthesizes a wide spectrum of biologically active substances [5,6,7]. These substances may exhibit both antagonistic and synergistic effects in relation to the neuron that synthesizes them as well as to neighuboring neurons. Therefore, they can enhance or inhibit the secretion of the main neurotransmitter. So, for example previous studies have shown that CART released by nitrergic neurons increases the nitric oxide contractile activity [44]. Therefore, in our studies, the method of double immunofluorescence staining was used to determine which substances are secreted in NOS-positive neurons and whether hyperglycaemia causes the same quantitative changes in relation to these substances as to the NOS. The results obtained during this experiment have shown co-localization of NOS with VIP, GAL and SP. Vasoactive intestinal polypeptide is regarded as one of the most important peptides associated with the intestinal regulatory processes. Primarily, VIP in the enteric neurons acts as an inhibitory factor, induces the smooth muscles diastole and down-regulates gastric acid secretion [5,46]. In the present study, we noted an increase in VIP-immunoreactivity inside the NOS-containing enteric neurons, both in the myenteric and submucosal plexuses. Augmentation of VIP in all segments of the small bowel may be a result of ant-inflammatory and/or neuroprotective action of VIP [47,48]. It should be noted that inflammation occurs in diabetes as a consequence of glucose toxicity [40]. Moreover, the previous study also reported an increase in VIP-LI enteric neurons in response to inflammatory processes [49]. Furthermore, increased VIP- immunoreactivity in enteric neurons with a simultaneous decrease in NOS expression may protect neurons from neuronal death. In turn, GAL, whose increased expression also been noted, may play a multidirectional function in the GI tract. The character of these functions clearly depends on the fragment of the digestive tract and the animal species studied. For instance, it is recognized that GAL up-regulates muscle contraction within the ileum of rat, guinea pig, rabbit and pig [44], whereas in the stomach and ileum of the dog, it exhibits antagonistic action [50]. To date, an increase in galanin immunoreactivity has been observed in the colon of non-obese diabetic mice and the ileum of 12-week-old diabetic rats [51]. The final substance whose expression was studied in this investigation was SP. We observed an increase in the population of neurons containing SP, in both the submucosal and myenteric plexuses. Only neurons inside the inner submucosal plexus in the duodenum did not show statistically significant changes. This observation strongly suggests that SP may be actively involved in the pathophysiological mechanisms of visceral pain during diabetes. This function seems to be very important because pain episodes are one of the most common disturbances in patients with long-term diabetes [2]. Moreover, the inflammatory and immune modulatory actions of SP are relatively well known [52,53]. Namely, SP is an important proinflammatory factor, which induces cytokine release [53], as well as an agent sending information between the nervous and immune systems [53]. This confirms that NOS-containing neurons through an increase in SP synthesis may be simultaneously involved in the regulation of pain stimuli and may modulate inflammatory response. 

Our results showed that nNOS containing neurons simultaneously express VIP, GAL and SP. However, the amount of which was dependent on the ganglion as well as the region of the small intestine. It should be emphasized that the hyperglycaemic resulted in an increase in the investigated substances in the submucosal and myenteric ganglia with a simultaneous decrease in the number of nNOS positive neurons. Considering that nitric oxide possesses neuroprotective activity, reduction of nitrergic neurons may create unfavourable conditions for the survival of enteric neurons. One of the defense mechanisms may be the increased of VIP and GAL expression. Previous in vitro studies conducted on isolated neurons from myenteric ganglion have shown that VIP increases the survival of neurons exposed to apoptotic factors [47]. GAL has also neuroprotective properties [53]. Therefore, it should be noted that as a result of impaired nitrergic transmission, the increase in the number of GAL and VIP positive neurons at least partially protects neurons from damage and possible cell death. In contrast, SP, which expression in nNOS neurons also increased, may indicate involvement of these neurons in sensory transmission. Full confirmation of the function of these substances requires further research, including the study of receptors or immunological mechanisms.

## 4. Materials and Methods

Ten juvenile female pigs of the White Large Polish breed, weighing 17 to 20 kg were used in the experiment. After acclimatization directly before diabetes induction the animals were divided into two groups: the diabetic group (D, *n* = 5) and the control group (C, *n* = 5). The treatment of animals was conducted in compliance with the instructions of the Local Ethical Committee in Olsztyn (Poland) (decision number 13/2015/DTN) and according to Act for the Protection of Animals for Scientific or Educational Purposes of 15 January 2015 (Official Gazette 2015, No. 266), applicable in the Republic of Poland with special attention paid to minimizing any stress reaction. After a week, diabetes was induced as previously described [52,53]. Streptozotocin (STZ) (Sigma-Aldrich, St Louis, MO, USA, S0130), 150 mg/kg of body weight, was dissolved in a freshly prepared disodium citrate buffer solution (pH = 4.23, 1 g streptozotocin/10 mL solution). Animals were anesthetized and the streptozotocin was injected via an intravenous needle inserted into an ear with ongoing infusion for roughly 5 min. In order to avoid gastrointestinal complications, mainly nausea and vomiting after diabetes induction, animals were fasted for 18 h before the experiment. The control pigs were injected with equal amounts of vehicle (citrate buffer). It is well known that streptozotocin can cause serious episodes of hyperglycaemia, 250 mL of 50% glucose solution per animal was given. After inducing diabetes, the animals were kept under standard laboratory conditions. They were fed standard fodder and had free access to water. The blood glucose concentration was estimated using an Accent-200 (Germany) biochemical analyser, with the colorimetric measurement at a wavelength of 510 nm/670 nm. For this aim, capillary blood from the ear was collected. The plasma glucose level was measured prior to the experiment beginning in both control and experimental groups. Next measurement was made 48 h after the injection of streptozotocin. Subsequent measurements of glucose levels were measured weekly until the end of the experiment.

Six weeks after streptozotocin injection, pigs were anesthetized via intravenous administration of pentobarbital (Vetbutal, Biowet, Poland) and perfused transcardially via the ascending aorta with freshly prepared 4% paraformaldehyde in 0.1 M (molar) phosphate buffer (pH 7.4). After perfusion, the small intestines were removed. Approximately 2 cm fragments of duodenum, jejunum and ileum were postfixed by immersion in the same fixative for 10 min, then washed with 0.1 M PB (pH 7.4) over 2 days and finally transferred and stored at 40 °C in an 18% buffered sucrose solution (pH 7.4), containing 0.001% natrium azide. The tissues were then kept at −80 °C until further processing. Frozen samples were cut in a cryostat (Microm HM 560 cryostat (Carl Zeiss, Jena, Germany) into 12-µm-thick sections and mounted on chrome alum-coated slides. Sections were processed by applying the routine double immunofluorescence technique. After drying at 32 °C for 45 min, the sections were rinsed in a phosphate buffer containing 0.8% sodium chloride and 0.02% potassium chloride (PBS, 3 × 10 min.) and incubated in 10% normal goat serum in PBS with 0.3% Triton X-100 (Sigma, USA) and 1% bovine serum albumin (BSA; Sigma, USA) for 20 min. The sections were then incubated overnight at 4 °C with primary antibodies diluted in PBS containing 0.3% Triton X-100 and 1% BSA raised against raised against Hu C/D (mouse polyclonal, Invitrogen, Gaithersburg, MD, USA; Cat # A-212711:1.000; working dilution 1:1000) and/or, nNOS (rabbit polyclonal, MercMillipore, Billerica, MA, USA; Cat # AB 5380; working dilution 1:6000), nNOS (mouse monoclonal, Sigma Aldrich, USA; Cat # N218; working dilatation 1:2000), GAL (rabbit polyclonal, MercMillipore. USA; Cat. # AB 2233; working dilution 1:2000), SP (rat monoclonal, AbD Serotec, Raleigh, NC, USA; Cat. # 8450-0505; working dilution 1:150), VIP (rabbit polyclonal, Biomol, Hamburg, Germany; Cat # VA1285; working dilution 1:5000). On the following day, the sections were rinsed (PBS, 5 × 15 min) and incubated with secondary antibodies (in PBS containing 0.25% BSA and 0.1% Triton X-100) for 4 h (Alexa Fluor 488 nm donkey anti-mouse, Invitrogen; Cat # A21202 working dilatation; 1:1000 and Alexa Fluor 546 nm goat anti rabbit, Invitrogen; Cat # A11010; working dilution 1:1000) to visualize the antibody combinations: Hu C/D /nNOS, nNOS/VIP; nNOS/GAL; nNOS/SP. The sections were then rinsed (PBS, 3 × 5 min) and covered with polyethylene glycol/glycerin solution containing DABCO (Sigma,). Standard controls, i.e., preabsorption for the neuropeptide antisera (20 μg of appropriate antigen per 1 mL of corresponding antibody at working dilution, as well as omission and replacement of the respective primary antiserum with the corresponding non-immune serum completely abolished immunofluorescence and eliminated specific staining. 

Using an Olympus BX51 microscope (Olympus, Hamburg, Germany) equipped with epi-illumination fluorescence filters immunostained neurons were analysed. Photographed were taken by a digital monochromatic camera (Olympus XM 10). The microscope was equipped with cellSens Dimension Image Processing software (Olympus,). For determination of the percentage of nNOS-LI neurons, at least 500 perikarya with clearly visible nucleus immunoreactive to Hu C/D in the particular type of enteric plexuses from each animal were investigated for the occurrence of nNOS. The obtained results were pooled and presented as mean ± SEM. To avoid double counting of the same perikarya, the investigated sections of intestine were located at least 100 μm apart. For the investigation, the co-localization of nNOS with other peptides, at least 150 nNOS-positive perikarya in particular types of enteric ganglia were investigated for immunoreactivity to particular neuronal factors. In these studies, nNOS-positive neurons were considered as representing 100%. The data pooled from all animal groups were statistically analysed using Statistica 13 software (StatSoft Inc., Tulsa, OK, USA) and expressed as a mean ± standard error (SEM) of mean. Significant differences were evaluated using Student’s *t*-test for independent samples (* *p* < 0.05, ** *p* < 0.01, and *** *p* < 0.001).

## 5. Conclusions

In conclusion, streptozotocin-induced diabetic pigs exhibit a significant decrease in NOS-positive submucosal and myenteric neurons in the small intestine. This is associated with an increase in VIP-, GAL- and SP- immunoreactivity in NOS-containing neurons. These observations suggest that chronic hyperglycaemia impairs the ability of neurons to synthesize nitric oxide. An increase in the expression of the above-mentioned substances in NOS-positive neurons (which have neuroprotective action) was simultaneously observed, may testify to the adaptation of neurons to oxidative stress and inflammatory conditions and reflects the well-known phenomenon of neuroplasticity. Moreover, selective noticeable loss of NOS-containing neurons alters gastrointestinal motility, leading to severe disturbances, such as heartburn, gastroparesis, diarrhoea etc. Unfortunately, the above-mentioned disturbances significantly reduce the quality of life of patients with diabetes. The treatment of these side effects is mainly symptomatic. Despite numerous studies on the diabetic gastroenteropathy, this issue is still insufficiently understood. Moreover, it is important to find an appropriate animal model. The use of pigs as a model for diabetic gastroenteropathy research seems to be better than the rodent model. Additionally it seems that enteric neuropeptides are involved in the gastric function and undergo changes during diabetes. This fact can be useful for developing a prevention strategy in patients with diabetes. 

## Figures and Tables

**Figure 1 ijms-20-01681-f001:**
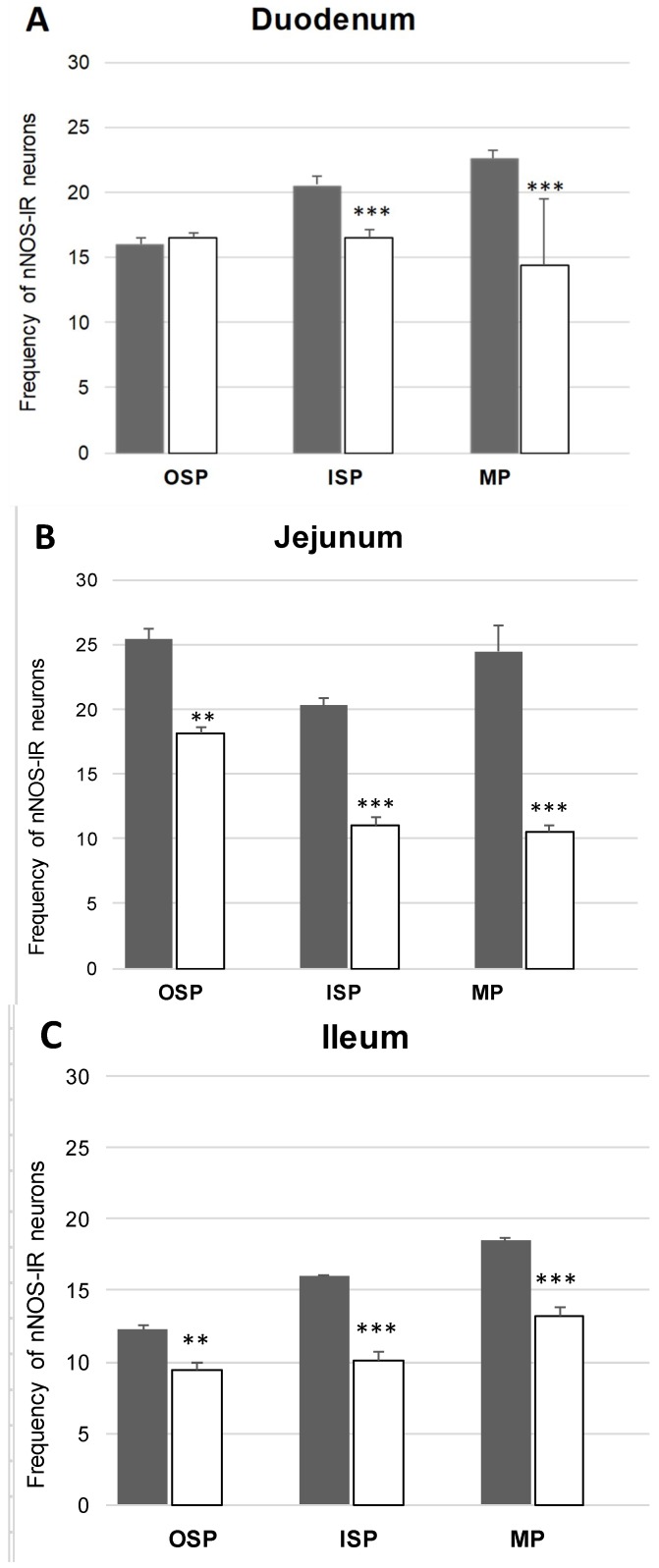
Neuronal nitric oxide synthase (nNOS)- positive cell bodies in inner submucous plexus (ISP), outer submucous plexus (OSP) and myenteric plexus (MP) within the duodenum, jejunum and ileum in the control and experimental groups. Mean (± SEM) percentage of neurons immunoreactive to nNOS in the OSP- intestinal outer submucous plexus, ISP- intestinal inner submucous plexus, MP- myenteric plexus in the duodenum (**A**), jejunum (**B**) and ileum (**C**). Control (grey bars), experimental (white bars), ** *p* < 0.01, *** *p* < 0.001—indicate differences between control and experimental group.

**Figure 2 ijms-20-01681-f002:**
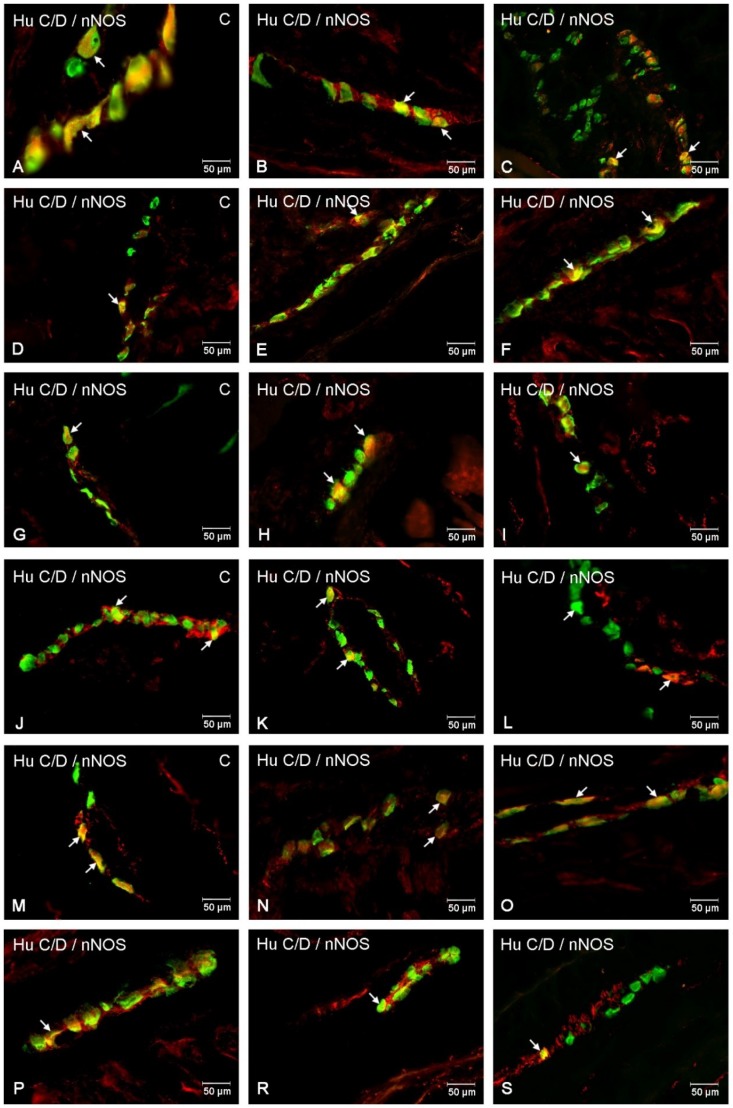
Intramural plexuses in the porcine duodenum, jejunum and ileum immunoreactive to nNOS. Myenteric plexus in the duodenum (**A**,**D**), jejunum (**B**,**E**) and ileum (**C**,**F**) under physiological conditions (A–C) and after streptozotocin administration (D–F). All photographs have been created by digital superimposition of two-colour channels (green for Hu C/D- used here as a pan-neuronal marker and red for nNOS). Neurons showing co-localization of Hu C/D and nNOS are indicated with arrows. Outer submucosal plexus in the duodenum (**G**,**J**), jejunum (**H**,**K**) and ileum (**I**,**L**) under physiological conditions (G–I) and after streptozotocin administration (J–L). All photographs were created by digital superimposition of two-colour channels (green for Hu C/D- used here as a pan-neuronal marker and red for nNOS). Neurons showing co-localization of Hu C/D and nNOS are indicated with arrows. Inner submucosal plexus in the duodenum (**M**,**P**), jejunum (**N**,**R**) and ileum (**O**,**S**) under physiological conditions (M–O) and after streptozotocin administration (P–S). All photographs were created by digital superimposition of two-colour channels (green for Hu C/D- used here as a pan-neuronal marker and red for nNOS). Neurons showing co-localization of Hu C/D and nNOS are indicated with arrows.

**Figure 3 ijms-20-01681-f003:**
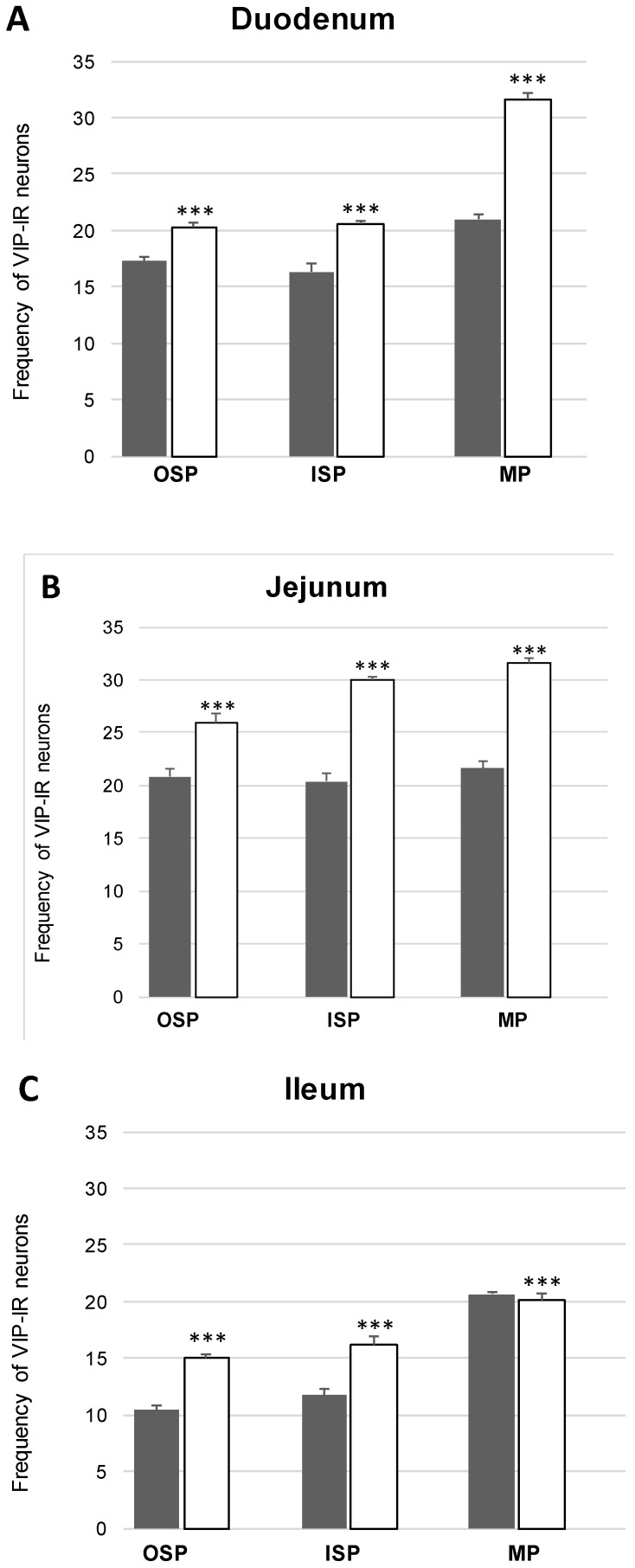
Colocalization of nNOS- positive cell bodies with VIP in inner submucous plexus (ISP), outer submucous plexus (OSP) and myenteric plexus (MP) with vasoactive intestinal polypeptide (VIP) within the duodenum, jejunum and ileum in the control and experimental groups. Mean (± SEM) percentage of neurons immunoreactive to n NOS and VIP in the MP- myenteric plexus, OSP- intestinal outer submucous plexus, ISP- intestinal inner submucous plexus in the duodenum (**A**), jejunum (**B**) and ileum (**C**). Control (grey bars), experimental (white bars), *** *p* < 0.001—indicate differences between control and experimental group.

**Figure 4 ijms-20-01681-f004:**
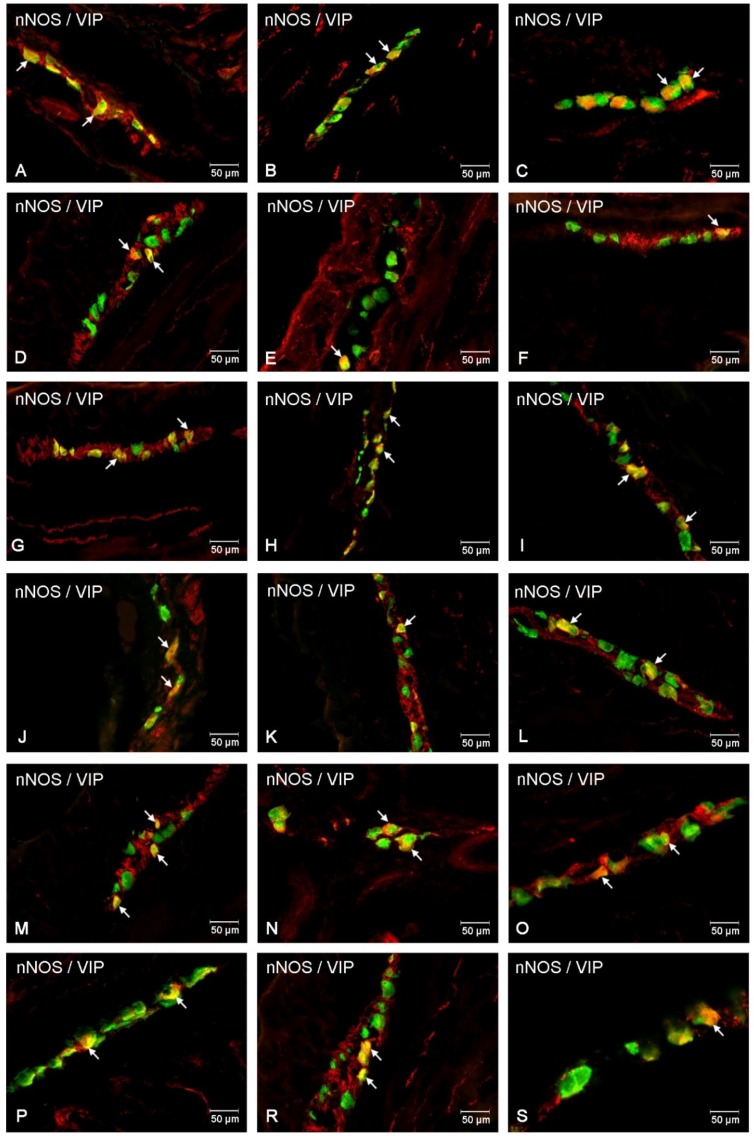
Intramural plexuses in the duodenum, jejunum and ileum immunoreactive to nNOS and VIP. Myenteric plexus in the duodenum (**A**,**D**), jejunum (**B**,**E**) and ileum (**C**,**F**) under physiological conditions (A–C) and after streptozotocin administration (D–F). All photographs were created by digital superimposition of two-colour channels (green for nNOS and red for VIP). Neurons showing co-localization of nNOS and VIP are indicated with arrows. Outer submucosal plexus in the duodenum (**G**,**J**), jejunum (**H**,**K**) and ileum (**I**,**L**) under physiological conditions (G–I) and after streptozotocin administration (J–L). All photographs have been created by digital superimposition of two-colour channels (green for nNOS and red for VIP). Neurons showing co-localization of nNOS and VIP are indicated with arrows. Inner submucosal plexus in the duodenum (**M**,**P**), jejunum (**N**,**R**) and ileum (**O**,**S**) under physiological conditions (M–O) and after streptozotocin administration (P–S). All photographs were created by digital superimposition of two-colour channels (green for nNOS and red for VIP). Neurons showing co-localization of nNOS and VIP are indicated with arrows.

**Figure 5 ijms-20-01681-f005:**
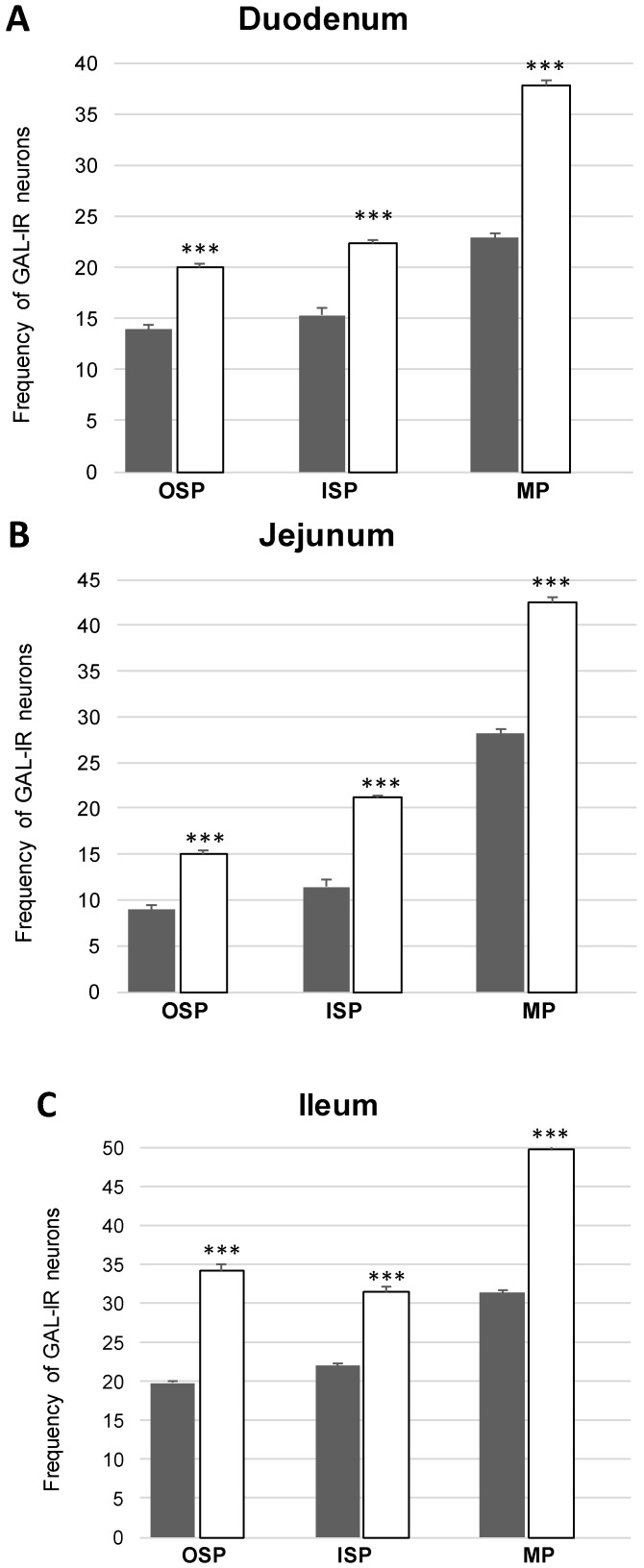
Colocalization of nNOS- positive cell bodies with GAL in inner submucous plexus (ISP), outer submucous plexus (OSP) and myenteric plexus (MP) with galanin (GAL) within the duodenum, jejunum and ileum in the control and experimental groups. Mean (± SEM) percentage of neurons immunoreactive to nNOS and GAL in the MP- myenteric plexus, OSP- intestinal outer submucous plexus, ISP- intestinal inner submucous plexus in the duodenum (**A**), jejunum (**B**) and ileum (**C**). Control (grey bars), experimental (white bars), *** *p* < 0.001—indicate differences between control and experimental group.

**Figure 6 ijms-20-01681-f006:**
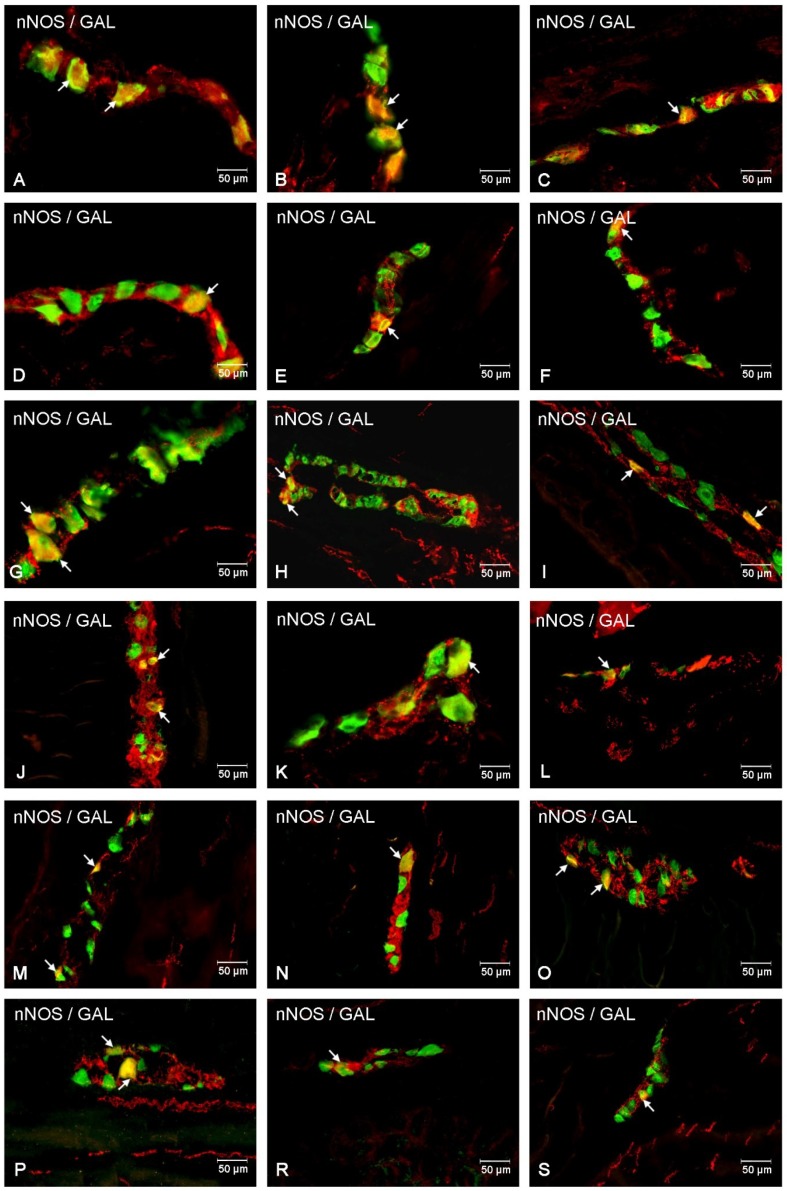
Intramural plexuses in the duodenum, jejunum and ileum immunoreactive to nNOS and GAL. Myenteric plexus in the duodenum (**A**,**D**), jejunum (**B**,**E**) and ileum (**C**,**F**) under physiological conditions (A–C) and after streptozotocin administration (D–F). All photographs were created by digital superimposition of two-colour channels (green for nNOS and red for GAL). Neurons showing co-localization of nNOS and GAL are indicated with arrows. Outer submucosal plexus in the duodenum (**G**,**J**), jejunum (**H**,**K**) and ileum (**I**,**L**) under physiological conditions (G–I) and after streptozotocin administration (J–L). All photographs have been created by digital superimposition of two-colour channels (green for nNOS and red for GAL). Neurons showing co-localization nNOS and GAL are indicated with arrows. Inner submucosal plexus in the duodenum (**M**,**P**), jejunum (**N**,**R**) and ileum (**O**,**S**) under physiological conditions (M–O) and after streptozotocin administration (P–S). All photographs were created by digital superimposition of two-colour channels (green for nNOS and red for GAL). Neurons showing co-localization of nNOS and GAL are indicated with arrows.

**Figure 7 ijms-20-01681-f007:**
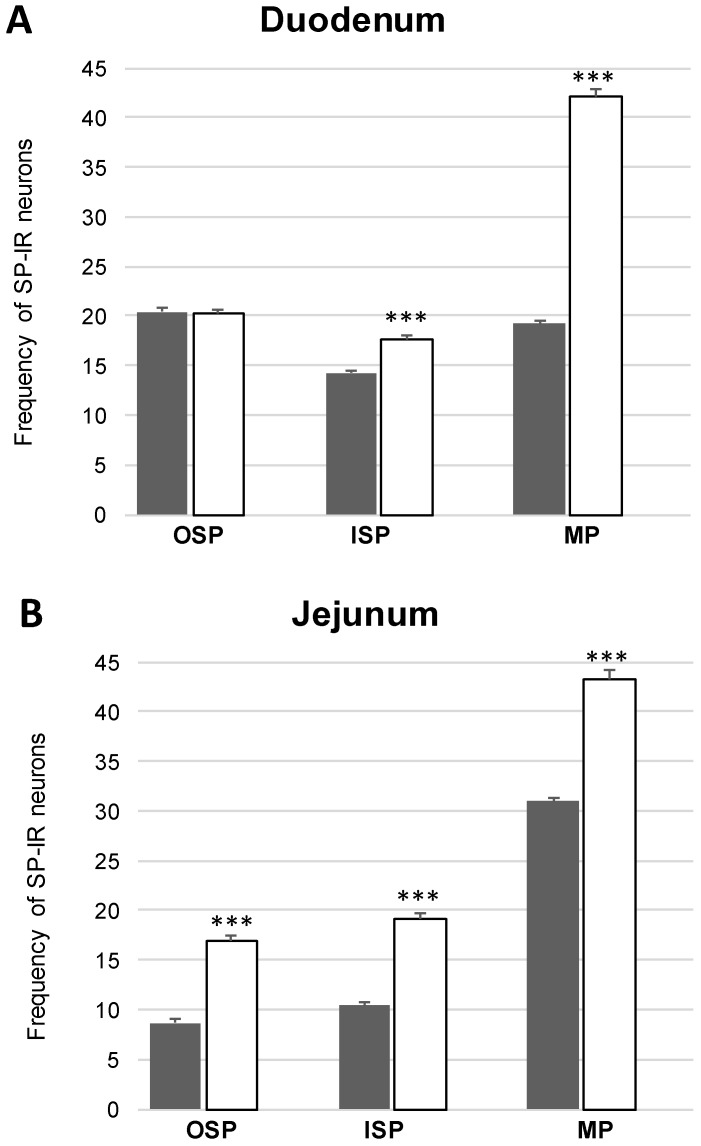
Colocalization of nNOS- positive cell bodies in the inner submucous plexus (ISP), outer submucous plexus (OSP) and myenteric plexus (MP) with substance P (SP) within the duodenum, jejunum and ileum in the control and experimental group. Mean (± SEM) percentage of neurons immunoreactive to nNOS and SP in the MP- myenteric plexus, OSP- intestinal outer submucous plexus, ISP- intestinal inner submucous plexus in the duodenum (**A**), jejunum (**B**) and ileum (**C**). Control (grey bars), experimental (white bars), ** *p* < 0.01, *** *p* < 0.001—indicate differences between control and experimental group.

**Figure 8 ijms-20-01681-f008:**
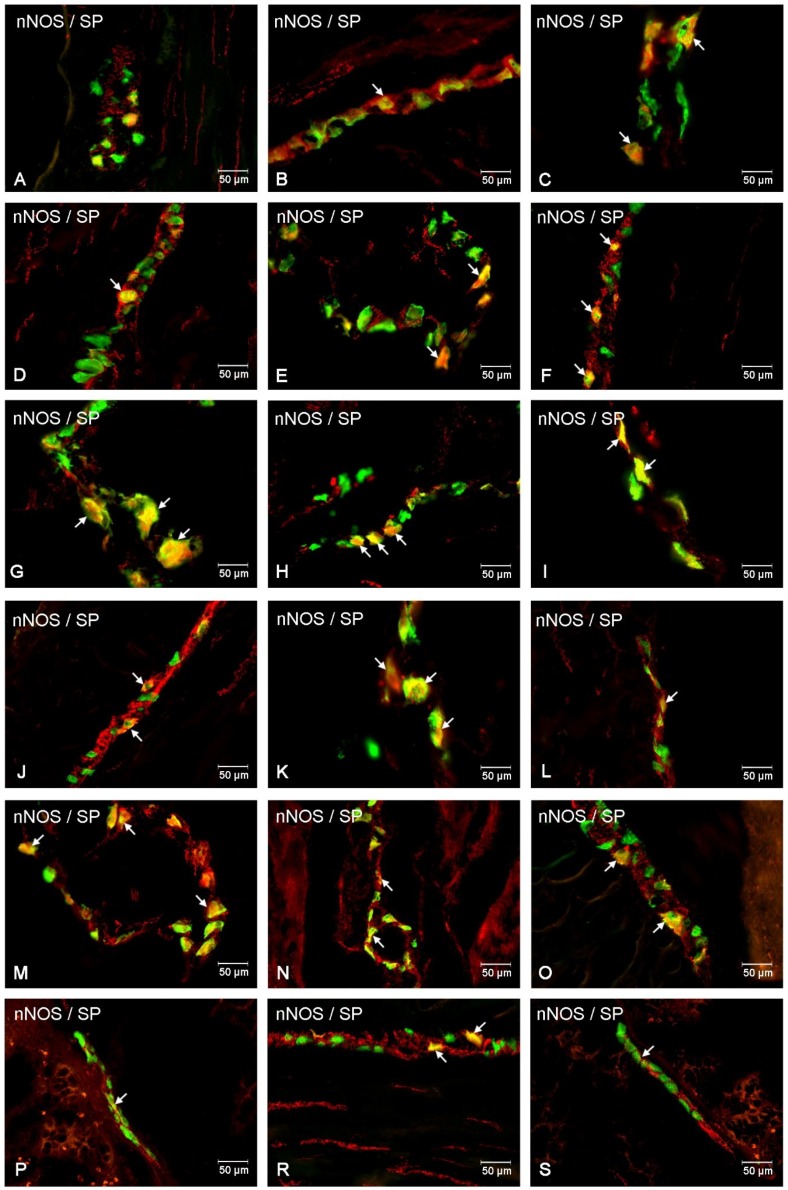
Intramural plexuses in the duodenum, jejunum and ileum immunoreactive to nNOS and SP. Myenteric plexus in the duodenum (**A**,**D**), jejunum (**B**,**E**) and ileum (**C**,**F**) under physiological condition (A–C) and after streptozotocin administration (D–F). All photographs were created by digital superimposition of two-colour channels (green for nNOS and red for SP). Neurons showing co-localization of nNOS and SP are indicated with arrows. Outer submucosal plexus in the duodenum (**G**,**J**), jejunum (**H**,**K**) and ileum (**I**,**L**) under physiological condition (G–I) and after streptozotocin administration (J–L). All photographs have been created by digital superimposition of two-colour channels (green for nNOS and red for SP). Neurons showing co-localization of nNOS and SP are indicated with arrows. Inner submucosal plexus in the duodenum (**M**,**P**), jejunum (**N**,**R**) and ileum (**O**,**S**) under physiological condition (M–O) and after streptozotocin administration (P–S). All photographs were created by digital superimposition of two-colour channels (green for nNOS and red for SP). Neurons showing co-localization of nNOS and SP are indicated with arrows.

**Table 1 ijms-20-01681-t001:** Serum glucose levels after induction of diabetes and glucose concentration after streptozotocin administration (up to six weeks).

Date of Blood Collection	Control Group (mmol/L)	SEM±	Experimental Group (mmol/L)	SEM±
Before streptozotocin injection	5.01	0.10	5.03	0.10
1 week after streptozotocin injection	5.08	0.10	17.36	0.38
2 weeks after streptozotocin injection	4.91	0.18	20.72	0.24
3 weeks after streptozotocin injection	5.19	0.06	21.58	0.27
4 weeks after streptozotocin injection	5.31	0.12	20.08	0.09
5 weeks after streptozotocin injection	4.84	0.32	22.26	1.21
6 weeks after streptozotocin injection	5.20	0.1	21.45	1.11

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
