# Peer review of "Hyperglycaemia-Induced Downregulation in Expression of nNOS Intramural Neurons of the Small Intestine in the Pig"

_ijms, 2019, doi:10.3390/ijms20071681_

Reviewer 1 Report

The publication submitted for review concerns the impact of hyperglycaemia on the level of nNOS and the degree of co-localization of nNOS and VIP, GAL and SP in small intestine neurons in pigs. Among the numerous vascular complications of diabetes, the pathomechanism of gastrointestinal autonomic neuropathy is the least well-known. Therefore, the results presented in the publication provide important knowledge about this disease. However,  I found many flaws that require authors' comment.  Please refer to the following comments:

Abstract - Abstract needs improvement, the objective of the work does not correspond to the presented results. It is not indicated why the degree of co-localization of nNOS and VIP, GAL and SP was assessed. The conclusion is very general. Authors should try to give the clinical significance of obtained results.

Introduction -  Introduction  is too long, especially the first paragraph. The order of the presented information should be changed, starting from the relationship between diabetes and GI autonomic neuropathy, the role of nNOS, VIPA, GAL and SP, then a brief description of enteric nervous system and finally justification of undertaking research.

Results and discussion. The authors employed only one research method, which is a serious disadvantage of this publication, especially in the case of attempts to find mechanisms responsible for obtained results. The authors suggest an important role of hyperglycaemia induced oxidative stress and inflammation in changes of the level of nNOS and the degree of co-localization. However, they have not provided any evidence to support these suggestions. It is worth to consider perfoming additional experiments, such as assessing the level of oxidative stress markers (MDA, 8-oxoG, etc.) and inflammatory  markers (CRP, TNFalpa, etc.) to provide mechanistic explanation.

The way the results are presented is unclear. Figures show pictures, however their description in the text contains  percentage and SD or SEM. What is the source of this data. Please add quantitative data (in the form of histohgrams) to the corresponding photos.

In the legend of each figure, the same description concernin co-localization is repeated three times.

The numerical and statistical data were  included in the tables. However, no information is provided about the type of data  (mean, median) or unit in which they were given. For example Data are expressed as mean +/- SD.

Neither in the Introduction nor in the Discussion Authors provided why they performed nNOS and VIP, GAL and SP co-localization experiments.

Please use glycemia or glucose level, the term glycemic level is confusing

The following lines142-151; 192-199; 239-246; 287-294 should be included to the legend of figures.

Materials and methods section. Line 412 - Authors omitted reference number.

There are numerous grammar, stylistic and editorial mistakes. The work requires extensive language edition. Here is a list of some examples of language errors:line  59-60 ...is considered as specific marker; line 125 ... the biggest drop...;no spaces line 430

Please explain the information presented in lines 372-373 (ref. 46)

Authors did not discuss the results concenrinig co-localization of nNOS and VIP, GAL and SP.o

There is no clinical significance of obtained results both in the discussion and conclusion.

Author Response

Editor-in-Chief

You will find included corrected version of our manuscript entitled “Hyperglycemia- induced down-regulation in expression of nNOS intramural neurons of the small intestine in the pig

” - Michał Bulc, Katarzyna Palus, Michał Dąbrowski, Jarosław Całka. We appreciate the thorough review. All text improvements of our manuscript have been done in red font.

Here are correction:

Comments from the editors and reviewers:
-Reviewer 1

1.    Abstract - Abstract needs improvement, the objective of the work does not correspond to the presented results. It is not indicated why the degree of co-localization of nNOS and VIP, GAL and SP was assessed. The conclusion is very general. Authors should try to give the clinical significance of obtained results.

Authors answer:

Thank you for your comment. As suggested by the referee abstract has been revised and improves according to referee suggestions.

2.   Introduction -  Introduction  is too long, especially the first paragraph. The order of the presented information should be changed, starting from the relationship between diabetes and GI autonomic neuropathy, the role of nNOS, VIPA, GAL and SP, then a brief description of enteric nervous system and finally justification of undertaking research.

Authors answer: Thank you for your comment. The introduction will be shortened as well as rewritten according to reviewer suggestion. The order of the paragraphs will be changed and the information about clinical significance will be added.

3.   Results and discussion. The authors employed only one research method, which is a serious disadvantage of this publication, especially in the case of attempts to find mechanisms responsible for obtained results. The authors suggest an important role of hyperglycaemia induced oxidative stress and inflammation in changes of the level of nNOS and the degree of co-localization. However, they have not provided any evidence to support these suggestions. It is worth to consider perfoming additional experiments, such as assessing the level of oxidative stress markers (MDA, 8-oxoG, etc.) and inflammatory  markers (CRP, TNFalpa, etc.) to provide mechanistic explanation.

Authors answer:

Thank you for your comment. The results presented in the present work were carried out as part of a scientific project focused on understanding changes in the chemical coding of enteric gastrointestinal neurons in response to chronic hyperglycaemia. The obtained results confirmed our assumptions about the neuroplasticity of enteric neurons and changes in the expression of neuroactive substances studied. Therefore, in our next research project (which are during application) we want to know more precisely the mechanism of gastrointestinal disorders in diabetes. These studies will include the contraction of selected parts of GI and the participation of advanced glycation end products and its receptor in the mechanisms of GI disorders in the course of diabetes.

4.  The way the results are presented is unclear. Figures show pictures, however their description in the text contains  percentage and SD or SEM. What is the source of this data. Please add quantitative data (in the form of histohgrams) to the corresponding photos.

Authors answer:

Thank you for your comment. For better transparency of the presented results tables will be converted into histograms.

5.   In the legend of each figure, the same description concernin co-localization is repeated three times.

Authors answer: The figures presented photographs of colocalization of investigated substances. Figure 1 presented neurons showing co-localization of Hu C/D (pan neuronal marker) and nNOS. Figure 2 neurons showing co-localization nNOS and VIP, in turn Figure 3 neurons showing co-localization nNOS and GAL and finally Figure 4 neurons showing co-localization nNOS and SP. Each of the figures presents a co-localization nNOS with other substances, which is detailed in the first sentence, while the other descriptions are the same because they concern the same segments of the gastrointestinal tract and the same plexuses.

6. The numerical and statistical data were  included in the tables. However, no information is provided about the type of data  (mean, median) or unit in which they were given. For example Data are expressed as mean +/- SD.

Authors answer:

As we mentioned in point 4, we will added histograms and new, more detailed descriptions.

7.   Neither in the Introduction nor in the Discussion Authors provided why they performed nNOS and VIP, GAL and SP co-localization experiments.

Authors answer:

This is right attention, we filled the missing information describing our choice of neuroactive substances to double immunofluorescence staining. Chapter discussion line 530-539.

8.   Please use glycemia or glucose level, the term glycemic level is confusing

Authors answer:

Thank you for your comment.  We improved this in our manuscript.

9.   The following lines142-151; 192-199; 239-246; 287-294 should be included to the legend of figures.

Authors answer:

Thank you for your comment. This editorial error was corrected.

10.  Materials and methods section. Line 412 - Authors omitted reference number.

Authors answer:

Thank you for your comment. The missing reference number was added. ,, Bulc, M.; Palus, K.; Całka J.; Zielonka Ł. Changes in Immunoreactivity of Sensory Substances within the Enteric Nervous System of the Porcine Stomach during Experimentally Induced Diabetes. J Diabetes Res. 2018 Jul 24;2018:4735659.’’

11.  There are numerous grammar, stylistic and editorial mistakes. The work requires extensive language edition. Here is a list of some examples of language errors: line  59-60 ...is considered as specific marker; line 125 ... the biggest drop...;no spaces line 430

Authors answer:

In order to avoid grammar, stylistic and editorial mistakes in this work, manuscript were checked by the native spiker. In addition, the certificate confirming this correction was attached.

12.  Please explain the information presented in lines 372-373 (ref. 46)

Authors answer:

,,Pharmacological inhibition of NOS by a 3-day oral administration of L-NG-nitroarginine methyl ester results in delayed gastric emptying [45]. However, in the small bowel, a decrease in nitric-expressing neurons during diabetes leads to a decrease in phase 3, which results in the prolongation of migrating myoelectric complex cycle [46]”.

In this statement, the authors sought to cite literature data describing the contribution of nNOS to the motor activity function of intestines and demonstrated that pathological reduction of nNOS content in the gastrointestinal tract resulted in decreased of gut motility. For better understanding, this sentence was minimally modified.

,,Pharmacological inhibition of NOS by a 3-day oral administration of L-NG-nitroarginine methyl ester results in delayed gastric emptying [45]. However, In the small bowel, a decrease in nitric-expressing neurons during diabetes leads to a decrease in phase 3, which results in the  …

13.  Authors did not discuss the results concenrinig co-localization of nNOS and VIP, GAL and SP.

Authors answer:

Thank you for your comment. The discussion concerning co-localization of nNOS and VIP, GAL and SP was added. Line 567-580.

14.  There is no clinical significance of obtained results both in the discussion and conclusion

Authors answer:

The missing information about clinical significance of obtained results was added in the conclusion. Line 657-663.

 Reviewer 2 Report

It is now well established that nitric oxide (NO) has a leading role as an inhibitory neurotransmitter of peripheral non-adrenergic, non-cholinergic nerves. Peripheral nitrergic nerves have a widespread distribution, and are particularly important in that they produce relaxation of smooth muscle in the gastrointestinal tract (GI). Hyperglycemia irreversibly damages both neurons – central and peripheral. Previously it was shown that a diminution in (neuronal nitric oxide synthase) nNOS expression and activity in the myenteric plexus is associated with the delay in colonic transit appearing with advanced age [Takahashi et al., 2000]. Diabetes patients complain about numerous GI disturbances. The goal of the study was to provide novel data on the influence of streptozotocin (STZ)‐induced hyperglycemia on the nitrergic neurons in the enteric nervous system (ENS) of the porcine small intestine. The results showed that porcine enteric neurons located inside the small intestine after six weeks of permanent hyperglycemia induced by STZ injection exhibited variability in the number of nNOS positive neurons. Additionally, it was demonstrated for the first‐time co‐localization of nNOS with other biologically‐active substances in animals with high glucose levels. Overall this is a well-designed study and provides very interesting results.

I have found only some minor mistakes:

-There are some inaccuracies in the second affiliation – a lack of the name and street number.

-Correspondence to:… DMV?

-Abstract: VIP, GAL and SP should be explained

-Keywords: I suggest replacing “diabetes” with “hyperglycemia”, “small intestine” removing, “synthetase” replacing with “synthase”.

-Introduction:

o   Authors are not constant using the abbreviations, e.g. nitric oxide, gastrointestinal tract, nitric oxide synthase, streptozocin.

o   There is a lack of appropriate reference (line 96)

Author Response

Editor-in-Chief

Enclosed please find a corrected version of our manuscript entitled “Hyperglycemia- induced down-regulation in expression of nNOS intramural neurons of the small intestine in the pig

” - Michał Bulc, Katarzyna Palus, Michał Dąbrowski, Jarosław Całka. We appreciate the thorough review. All text improvements of our manuscript have been done in red font.

Here are correction:

Comments from the editors and reviewers:
Reviewer 2

1. There are some inaccuracies in the second affiliation – a lack of the name and street number.

-Correspondence to:… DMV?

-Abstract: VIP, GAL and SP should be explained

-Keywords: I suggest replacing “diabetes” with “hyperglycemia”, “small intestine” removing, “synthetase” replacing with “synthase”.

2. Introduction:

-  Authors are not constant using the abbreviations, e.g. nitric oxide, gastrointestinal tract, nitric oxide synthase, streptozocin.

Round  2

Reviewer 1 Report

Authors provided sufficient answers to almost all of my comments. However, I am a little bit confused with answers for remark 2 and 4. I am not sure whether there have been introduced changes in the manuscript (There are only two red sentences in the Introduction). Besides, the Authors wrote in response "The introduction will be shortened as well as rewritten according to reviewer suggestion. The order of the paragraphs will be changed and the information about clinical significance will be added."

There are histograms in the manuscript. Why the Authors wrote in answer for remark 4 "For better transparency of the presented results tables will be converted into histograms."

Author Response

Editor-in-Chief

Enclosed please find a corrected version of our manuscript entitled “Hyperglycemia- induced down-regulation in expression of nNOS intramural neurons of the small intestine in the pig

” - Michał Bulc, Katarzyna Palus, Michał Dąbrowski, Jarosław Całka. We appreciate the thorough review.

Here are correction:

Comments from the editors and reviewers:
Reviewer 1

Authors provided sufficient answers to almost all of my comments. However, I am a little bit confused with answers for remark 2 and 4. I am not sure whether there have been introduced changes in the manuscript (There are only two red sentences in the Introduction). Besides, the Authors wrote in response "The introduction will be shortened as well as rewritten according to reviewer suggestion. The order of the paragraphs will be changed and the information about clinical significance will be added."

Authors answer :

With regard to the introduction, we have changed presented information. In the first version of the introduction the first and second paragraphs described the organization of ENS. The following presented description of nNOS functions in ENS, next described gastrointestinal complications during diabetes and the last one showed the aim of the study. After revision we changed the order of the paragraphs as suggested by the reviewer. We started with the description of changes in gastrointestinal tract during diabetes, next we focused on the description of organization and functions of GI and the role of nNOS in GI. The last paragraphs include information about the goal of the study. Moreover some sentences were removed like the reviewer suggested, with one extra addition being that in the final version we decided to shorten the text by removing the following lines 73-75 ,, Although the physiological role of NO is well-known, many pathological aspects connected with the functions of NO in the enteric neurons are not fully understood.”. Clinical information was added in the abstract ,, This observation confirmed that diabetic hyperglycemia can cause changes in the neurochemical characteristics of enteric neurons, which can therefore lead to numerous disturbances in gastrointestinal tract functions. Moreover, can be the basis for elaboration of these peptides analogues utilized as therapeutic agents in treatment of GI complications’’ and in the introduction “Only full understanding of this mechanism and understanding the role of various biological factors including neurotransmitters will allow to conduct research on substances with a potential therapeutic effect”.

It is worth to underline that here the basic study is presented, not clinical trials. Therefore, all the practical use of the obtained results requires a lot of research and our clinical conclusions are cautious.

There are also histograms in the manuscript. The answer why the Authors wrote them is in remark 4: "For better transparency of the presented results tables will be converted into histograms.".

Authors answer :

The table contains a large number of data that cannot be always properly read. Histograms, however, enable quick analysis of results and show changes in the experimental group in a transparent way, as the reader has results listed side by side.